# Sequential AI-ECG Diagnostic Protocol for Opportunistic Atrial Fibrillation Screening: A Retrospective Single-Center Study

**DOI:** 10.3390/jcm14186675

**Published:** 2025-09-22

**Authors:** Ji-Hoon Choi, Sung-Hee Song, Jongwoo Kim, JaeHu Jeon, KyungChang Woo, Soo Jin Cho, Seung-Jung Park, Young Keun On, Ju Youn Kim, Kyoung-Min Park

**Affiliations:** 1Division of Cardiology, Department of Internal Medicine, Konkuk University Medical Center, Konkuk University School of Medicine, Seoul 05030, Republic of Korea; cardiol2p@gmail.com; 2Wellysis Corp., Seoul 06159, Republic of Korea; shea.song@wellysis.com (S.-H.S.); jongwoo.kim@wellysis.com (J.K.); 3MediFarmSoft Co., Ltd., Seoul 05836, Republic of Korea; ceo@medifarmsoft.com (J.J.); kc@medifarmsoft.com (K.W.); 4Center for Health Promotion, Samsung Medical Center, Sungkyunkwan University School of Medicine, Seoul 06351, Republic of Korea; soojin77.cho@samsung.com; 5Division of Cardiology, Department of Medicine, Heart Vascular Stroke Institute, Samsung Medical Center, Sungkyunkwan University School of Medicine, Seoul 06351, Republic of Korea; orthovics@gmail.com (S.-J.P.); yk.on@samsung.com (Y.K.O.); kzzoo921@gmail.com (J.Y.K.)

**Keywords:** atrial fibrillation, electrocardiogram, artificial intelligence, screening

## Abstract

**Background/Objectives**: Atrial fibrillation (AF) often occurs in episodes that are sudden and go unnoticed, reducing the chances of anticoagulation. We evaluated a two-stage AI ECG screening protocol that uses a single ECG model at initial screening and, if necessary, a serial ECG model after short interval follow-up to enhance accuracy while saving monitoring resources. **Methods**: We analyzed 248,612 12-lead ECGs from 164,793 adults (AF, *n* = 10,735) for model development and assessed the protocol in 11,349 eligible patients with longitudinal ECGs. The proposed algorithm first applied a single-ECG AI model at the initial visit, followed by a serial-ECG AI model three months later if AF was not initially detected. The model’s performance was evaluated using several metrics, including the area under the receiver operating characteristic curve (AUROC), sensitivity, specificity, accuracy, and F1 score. **Results**: The protocol achieved an AUROC of 0.908 with a sensitivity of 88.1%, specificity of 78.7%, positive predictive value (PPV) of 30.2%, negative predictive value (NPV) of 98.4%, accuracy of 79.6%, and an F1 score of 0.450. Among patients with a history of stroke (*n* = 551), 84.9% were correctly identified as AF-positive under the protocol. **Conclusions**: A sequential AI ECG strategy maintains high NPV at entry and improves PPV with longitudinal confirmation. This approach can prioritize ambulatory monitoring for those most likely to benefit and merits prospective, multi-center validation and cost-effectiveness assessment.

## 1. Introduction

Atrial fibrillation (AF) is the most common sustained arrhythmia and remains a leading cause of cardioembolic stroke [1,2]. Current guidelines promote proactive case detection and structured pathways to minimize missed AF cases and speed up prevention efforts, while recognizing uncertainties about the best screening methods and workflows [1]. Consumer wearables and single-lead devices have facilitated large-scale rhythm monitoring outside clinics, but population prevalence limits positive predictive value, so most programs still depend on confirmatory testing for diagnosis [3]. Patch-based ambulatory monitors and implantable loop recorders (ILR) improve detection, especially after cryptogenic stroke, but they require significant resources and cannot be used indiscriminately [4,5]. Consequently, many health systems face the challenge of deciding whom to monitor, when, and how often to reassess risk, rather than whether to screen at all.

Artificial intelligence (AI)–enabled analysis of standard 12-lead electrocardiograms (ECGs) recorded during sinus rhythm offers an alternative approach: instead of intermittently detecting AF itself, models can infer latent atrial cardiomyopathy and near-term AF risk from subtle features in sinus rhythm [6,7]. Early research shows that convolutional neural network models trained on routine 12-lead ECGs can identify individuals with AF (or imminent AF) even when the tracing appears normal at the time of care, suggesting that AI-ECG could serve as a pre-screening tool to prioritize further rhythm monitoring [6,7]. Since this method uses existing equipment and fits into standard outpatient visits, it can provide immediate, actionable risk assessments without requiring patient-owned devices or extended monitoring.

Building on this concept, we previously reported a single-ECG machine-learning model that estimated AF risk during sinus rhythm using a standard 12-lead tracing, demonstrating strong discrimination in routine clinical data [8]. Although single-timepoint estimation offers high negative predictive value, it may provide false reassurance if risk evolves over time; conversely, indiscriminate escalation based on one elevated score can trigger unnecessary monitoring in low-prevalence settings [1,3,4]. These limitations motivate a sequential strategy that explicitly accounts for temporal evolution of risk while conserving resources.

In this study, we assess a sequential, two-visit AI-ECG protocol that analyzes a baseline 12-lead ECG and a second ECG taken after a short interval, escalating to ambulatory monitoring only if risk remains persistently high. The staged approach aims to maintain sensitivity and negative predictive value at initial assessment—features that enable safe deferral in low-risk patients—while enhancing accuracy at follow-up by requiring long-term confirmation before allocating limited monitoring resources. From a healthcare system perspective, this triage could improve diagnostic accuracy and target monitoring to patients most likely to benefit, with potential downstream improvements in cost-effectiveness [9]. We hypothesized that the sequential design would preserve a high negative predictive value to safely defer monitoring in low-risk individuals while increasing detection among those with consistently high risk, especially in subgroups where timely AF detection could influence antithrombotic treatment.

The present work does not introduce a new algorithm; rather, it operationalizes a diagnostic protocol for opportunistic AF screening using models previously described in our prior study [8] and evaluates their performance and workflow implications in a clinic-embedded setting.

## 2. Method

### 2.1. Study Design and Oversight

This retrospective review of a diagnostic protocol was conducted at a tertiary academic hospital. The study included adults (≥18 years) with standard 12-lead ECGs recorded from 2010 to 2021. The Institutional Review Board (IRB) approved the study and granted a waiver of informed consent because de-identified data was used (IRB No. SMC 2020-01-007).

### 2.2. ECG Acquisition and Preprocessing

ECGs were recorded using Philips PageWriter systems (Philips Medical Systems, Andover, MA, USA) at a sampling rate of 500 Hz for 10 s, with a 5 μV resolution. Standard 12-lead electrode positions were used (Figure 1). Limb electrodes were placed on the right arm, left arm, right leg, and left leg (on distal limbs or, when necessary, proximally on the limbs to reduce artefact while preserving Einthoven geometry). Precordial electrodes were positioned at V1 at the 4th intercostal space at the right sternal border; V2 at the 4th intercostal space at the left sternal border; V4 at the 5th intercostal space on the mid-clavicular line; V3 midway between V2 and V4; V5 on the anterior axillary line at the same horizontal level as V4; and V6 on the mid-axillary line at the same level as V4. These placements were applied consistently across recordings.

Nine independent leads, excluding augmented limb leads, were processed. The signals were subjected to standard filtering and segmentation. From the P-QRS-T complexes, we extracted features at the beat, interval, and morphology levels. Additionally, for the serial model, delta features were derived by calculating differences between corresponding features in baseline and follow-up ECGs.

### 2.3. Outcome Definition and Adjudication

Patients were classified as having definite AF if AF was documented on a 12-lead ECG or Holter monitor and confirmed in their health records. The earliest confirmed AF date was set as the index AF date. The comparison group consisted of individuals without AF documentation but with adjudicated normal sinus rhythm (NSR) ECGs. Exclusion criteria included: (i) a prior AF diagnosis before the index NSR ECG; (ii) no NSR ECG before the index date; (iii) only one NSR ECG available; (iv) an AF indication without a positive AF ECG; (v) incomplete records preventing careful adjudication; and (vi) ECGs that could not be classified as NSR.

### 2.4. Cohorts and Splitting

For model development, we analyzed 248,612 ECGs from 164,793 patients (including 10,735 with AF). Data splits were performed at the patient level to prevent data leakage, with separate development (training/validation) and test sets. We also detail the calendar windows for each split and perform a temporal sensitivity analysis, where training data used earlier cases and testing used later cases. For temporal robustness, training/validation used earlier calendar windows, and the test set comprised a later window that was not accessed during development; all splits were at the patient level. The protocol evaluation set included 11,349 patients with longitudinal ECGs suitable for staged assessment; a subgroup with prior stroke (*n* = 551) was analyzed specifically for clinical significance.

### 2.5. Model Development

We developed a two-step AI-ECG workflow that maps engineered ECG features to the probability of near-term AF. A single-ECG classifier is applied at the baseline visit, and a serial-ECG classifier is applied at a short-interval follow-up. Both models were trained strictly on development partitions defined elsewhere in this Methods section and did not access any information from the test partition.

For input representation, the single-ECG model consumes a fixed-length vector of morphology and timing descriptors derived from sinus-rhythm 12-lead ECGs. The serial-ECG model uses the within-person change between two ECGs by taking the element-wise difference (follow-up minus baseline) of the same feature set, emphasizing temporal progression rather than absolute levels. Pair construction for the serial step, including the prespecified blanking rule, follows the study’s cohort definition described elsewhere.

The feature taxonomy included peak amplitudes, conventional intervals, segment levels, and durations, with beat-level quantities summarized by mean, minimum, maximum, and standard deviation. To reflect atrial remodeling signals, we also used compact P-wave shape indices, beat-wise correlation statistics against a per-record template, a narrow-band atrial-activity index, and concise heart-rate-variability summaries. Age and sex were appended as non-ECG covariates. Feature generation adhered to consistent conventions across all experiments.

Modeling was performed with gradient-boosted decision trees (LightGBM) using a logistic objective. We employed Bayesian optimization over a bounded hyperparameter space (number of estimators, learning rate, maximum depth/num_leaves, subsample, colsample_bytree, min_data_in_leaf, and class weights) with early stopping on the validation split to control overfitting. Class imbalance was handled via class weighting during training. All runs used fixed random seeds and version-locked dependencies to ensure determinism and auditability.

Model selection was based on validation of the area under the receiver operating characteristic curve (AUROC). Operating points used in the staged workflow were prespecified and are described in the thresholding subsection; the serial-stage operating point corresponded to Youden’s J, while the baseline stage used a sensitivity-oriented operating point appropriate for screening. Performance metrics and calibration procedures are detailed in the statistical evaluation subsection.

The primary model was a LightGBM optimized with a logistic loss. Because tree ensembles are not layer-based, notions such as “layers” and “activation functions” do not apply; model capacity and structure are governed by tree depth/number of leaves, learning rate (shrinkage), and the number of boosting iterations. Regularization included row/column subsampling, split constraints (e.g., min_data_in_leaf, minimum split gain), L1/L2 penalties on leaf weights, and early stopping on the patient-level validation split. Class weighting addressed imbalance.

For comprehensive algorithmic definitions, feature computation protocols, and additional implementation details, please refer to “Machine Learning Algorithm to Predict Atrial Fibrillation Using Serial 12-Lead ECGs Based on Left Atrial Remodeling” [8] and its Supplementary Material.

### 2.6. Threshold Selection and Decision Logic

On the validation set, we identified two operating points: (i) Youden’s J and (ii) a sensitivity-focused point in the 0.85–0.90 range suitable for screening. In the staged protocol, the baseline step (using a single-ECG model) employs the sensitivity-focused threshold to ensure a high NPV, while the follow-up step (with a serial-ECG model) uses Youden’s J to improve positive predictive value (PPV). Confusion matrices and scenario examples are included in the text to help interpretation.

### 2.7. Metrics and Statistical Analysis

We report AUROC with 95% confidence intervals (CIs), sensitivity, specificity, PPV/negative predictive value (NPV), accuracy, and F1 score. Continuous variables are summarized as mean ± standard deviation (SD) and analyzed with appropriate tests, while categorical variables are presented as counts and percentages. Bootstrap resampling was employed to calculate CIs when applicable. We additionally report precision–recall AUC and calibration (intercept and slope) with patient-level bootstrap 95% CIs to complement AUROC in this imbalanced setting.

### 2.8. Proposed Algorithm for Screening Atrial Fibrillation (Figure 2)

The proposed algorithm employs a sequential screening process to identify AF. It involves two main stages: an initial prediction using a single ECG model, followed by an assessment with a serial ECG model. During the first visit, a 12-lead ECG is recorded and analyzed with an AI-based model to estimate the likelihood of AF. If AF is detected at this point, the patient will undergo intensive ECG monitoring via wearable devices for further investigation. If not, a follow-up visit is scheduled after three months for reassessment. At this follow-up, a second 12-lead ECG is performed. The serial ECG model combines data from both ECGs to improve detection accuracy. If AF is predicted then, the patient is referred for intensive monitoring with wearable devices. If not, routine screening with a 12-lead ECG every six months continues.

This algorithm is designed to sequentially evaluate the risk of AF using both single and serial ECG analyses. Patients identified as having a higher likelihood of AF based on either model will undergo continuous monitoring, while those with negative results will continue with routine ECG assessments at regular intervals.

## 3. Results

### 3.1. Baseline Characteristics (Table 1) for Developing Two AI Models

A total of 2,083,335 ECGs were collected from 883,568 adult patients. Of these, 1,688,419 ECGs from 712,547 patients were excluded based on the study criteria. Ultimately, 164,793 patients took part, with 154,058 in the NSR group and 10,735 in the AF group. The AF group had a significantly higher mean age (65.7 ± 13.0 years) than the NSR group (56.5 ± 13.5 years; *p* < 0.001). Additionally, a larger proportion of males was observed in the AF group (53.9%) compared to the NSR group (46.9%; *p* < 0.001). Patients in the AF group also had more ECGs on average (3.5 ± 3.8) than those in the NSR group (2.1 ± 1.6; *p* < 0.001).

### 3.2. Performance of the Proposed Algorithm

The study involved 11,349 patients to assess the effectiveness of the proposed algorithm for AF screening. Participants were split into two groups: a Normal Group with 10,274 patients who had no history of AF, and an AF Group with 1075 patients diagnosed with AF. Of those in the AF Group, 551 had a history of embolic stroke, indicating a subset with cerebrovascular issues linked to AF.

The proposed AI-enabled sequential ECG screening algorithm was used in this study to evaluate its ability to predict AF cases. Its classification performance was assessed by comparing predictions with the actual clinical diagnoses of AF in the dataset. The algorithm achieved a sensitivity of 88.1%, showing a strong capacity to detect AF correctly, and a specificity of 78.7%, indicating its effectiveness in excluding non-AF cases. The PPV was 30.2%, meaning about one-third of patients predicted to have AF were true positives, while the NPV was 98.4%, demonstrating a high ability to rule out AF in negative cases. The overall accuracy of the model was 79.6%, and the F1 score was 0.450, which balances the precision and recall (Table 2). The confusion matrix (Figure 3) displays the model’s classification results. It correctly identified 8088 negative cases and 947 positive cases. However, 2186 negative cases were wrongly classified as positive, and 128 positive cases were misclassified as negative.

The model showed a strong ability to detect AF, with an AUROC of 0.908 (Figure 4). A subgroup analysis was performed for patients with a history of stroke (*n* = 551), a group especially relevant due to the close link between AF and thromboembolic events. The model identified 468 out of 551 stroke patients (84.9%) as having AF, demonstrating a high detection rate in this clinically important subgroup. These findings emphasize the algorithm’s effectiveness in screening for AF in the general population and its consistent performance in stroke patients, where early AF detection is vital for secondary prevention.

## 4. Discussion

### 4.1. Principal Findings

This study demonstrated the proposed AF screening protocol using a sequential ECG AI model. The AI-enabled sequential ECG screening algorithm achieved high accuracy (AUROC 0.908, sensitivity 88.1%) in detecting AF, particularly in high-stroke-risk patients, with an 84.9% detection rate. These results support the workflow design to safely exclude AF at baseline and escalate testing only when ongoing risk persists.

### 4.2. How This Work Relates to Prior Evidence

Our rationale is based directly on previous AI-ECG research, which shows that sinus-rhythm tracings contain hidden atrial remodeling linked to near-term AF risk. Attia et al. demonstrated that an AI-enabled ECG can identify individuals with AF despite normal-looking tracings, establishing proof of concept for “pre-AF” signal detection in routine 12-lead ECGs [7]. Extending this idea, our earlier research found that serial ECG features (i.e., changes within a patient over time) outperform single-timepoint features in predicting incident AF—this observation motivated the sequential confirmation step used here [8]. Systematic reviews and meta-analyses have also concluded that AI approaches can detect AF or AF risk with high accuracy, while highlighting the need for prospective outcome studies [10].

Our protocol is also complementary to wearable-based screening strategies. The Apple Heart Study demonstrated the scalability of photoplethysmography (PPG)-based AF notifications, albeit with recognized concerns about follow-up and potential over-referral [3]. Randomized data from mSToPS showed that patch monitoring increases AF detection and anticoagulation initiation but can increase overall resource utilization [5]. In this context, a clinic-embedded AI-ECG pre-screen that preserves NPV at the first visit and enriches PPV at a short follow-up visit may help triage who should proceed to ambulatory monitoring.

### 4.3. Alignment with Guidance and the Screening Debate

Current guidance underscores both the opportunity and uncertainty around AF screening. The 2024 ESC AF guidelines encourage systematic approaches to case finding and integrated pathways, while acknowledging gaps regarding ideal modalities and workflows [1]. In contrast, the USPSTF issued an “I” statement (insufficient evidence) for screening asymptomatic adults, highlighting ongoing equipoise about clinical benefit and potential harms from downstream testing [11]. Our staged protocol is designed to operate within these realities: it seeks to defer costly monitoring when baseline AI-ECG risk is low and to confirm persistence of risk before escalation, thereby mitigating false positives in low-prevalence populations.

### 4.4. Interpreting the False-Positive Burden and Threshold Selection

The PPV of 30.2% observed here is consistent with screening in low-prevalence settings and should be interpreted alongside very high NPV (98.4%). By prespecifying a sensitivity-targeted threshold at baseline and applying Youden’s J at the serial step, we prioritized safety first and precision second. This “preserve NPV, then recover PPV” approach echoes general principles of multi-stage diagnostics and can be further appraised with decision-curve analysis when prospective data become available [12]. Clinically, a modest PPV implies that a subset of patients will undergo unnecessary ambulatory monitoring; however, the serial-confirmation requirement is intended to reduce such referrals compared with a single-timepoint trigger, particularly among intermediate-risk individuals.

### 4.5. Clinical and Health-System Implications

From a workflow perspective, the protocol leverages standard 12-lead ECGs during routine visits without requiring patient-owned devices. When paired with clear triage rules and a short follow-up interval, this may focus ambulatory monitoring (patches, wearables, or ILRs) on those with persistently elevated risk. Trials and economic models suggest that targeted monitoring strategies can be cost-effective under plausible assumptions, although estimates vary with age, baseline risk, and follow-up intensity [9]. Our data therefore support using AI-ECG as a gatekeeper to downstream monitoring rather than a replacement for confirmatory rhythm documentation.

### 4.6. Comparison with Other AI Experiences in Electrophysiology

AI has been examined in electrophysiology beyond AF detection and prediction, including risk stratification, ablation planning, and post-procedure recurrence forecasting. Recent reviews cover its use with wearable PPG/ECG signals, 12-lead ECGs, and intracardiac data, highlighting the importance of interpretability and validation [10]. Although many narrative and exploratory studies show enthusiasm for AI in electrophysiology (EP) workflows, they also emphasize the need for transparent model reporting and calibration checks in clinical practice [13]. Our study offers a protocol-level perspective—focusing on how to utilize AI-ECG outputs for structuring care—rather than introducing a new network architecture.

### 4.7. Strengths and Limitations

Strengths include (i) the ability to evaluate in routine clinical ECGs across various care settings, (ii) a sequential design that aligns with screening principles, and (iii) transparent reporting of performance metrics, including PPV/NPV trade-offs. However, there are limitations to consider. First, the study is retrospective and conducted at a single center; outcomes such as stroke reduction and anticoagulation appropriateness were not examined, meaning no claims can be made about clinical benefits. Second, reliance on specific devices and pipelines may restrict generalizability; adapting the model for different ECG vendors and regular recalibration is necessary for broader use. Third, despite using patient-level splits, residual label misclassification may occur due to paroxysmal AF and incomplete monitoring, potentially biasing results toward null. Lastly, PPV is modest at the population level, so implementation should include clear follow-up procedures and auditing to monitor false-positive effects like benign symptom amplification and clinic congestion.

### 4.8. Future Directions

Prospective, multi-center pragmatic trials are needed to determine whether AI-ECG–guided pathways improve patient-important outcomes (stroke, bleeding, quality of life) and are cost-effective relative to usual care, opportunistic screening, or wearable-first strategies [1,9,11]. Trial designs should (i) randomize at the pathway level (AI-ECG gatekeeping vs. standard referral), (ii) predefine resource utilization endpoints, and (iii) incorporate decision-curve or net-benefit analyses to align operating points with clinically acceptable trade-offs [12,14]. Given signals that serial ECG information augments risk stratification, we also encourage investigations into optimal follow-up intervals and patient segments (e.g., prior stroke/transient ischemic attack, high CHA_2_DS_2_-VASc) most likely to benefit from the staged approach [8].

## 5. Conclusions

The proposed two-stage, AI-enabled sequential 12-lead ECG protocol appears to achieve strong discrimination in routine care, with high NPV at entry and potential gains in precision after serial confirmation—particularly among patients with prior stroke. Implemented as a gatekeeping step within existing workflows, it may help target ambulatory monitoring toward those with persistently elevated risk and could improve the efficiency of AF detection and resource use. Given the retrospective, single-center design and absence of outcome data, these findings should be interpreted prudently; prospective, multi-center evaluations—including integration with wearable ECG strategies and formal cost-effectiveness analyses—are needed to determine generalizability and possible impact on earlier AF identification and stroke prevention.

## Figures and Tables

**Figure 1 jcm-14-06675-f001:**
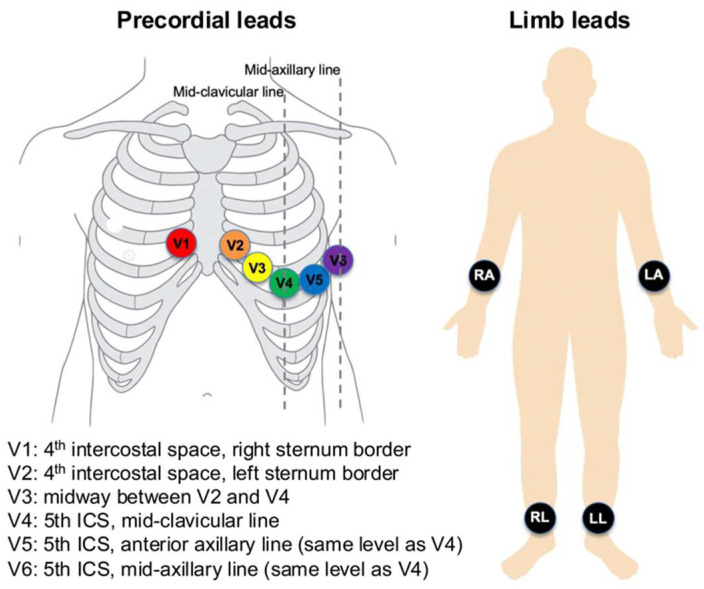
Standard 12-lead electrode placement (precordial and limb leads).

**Figure 2 jcm-14-06675-f002:**
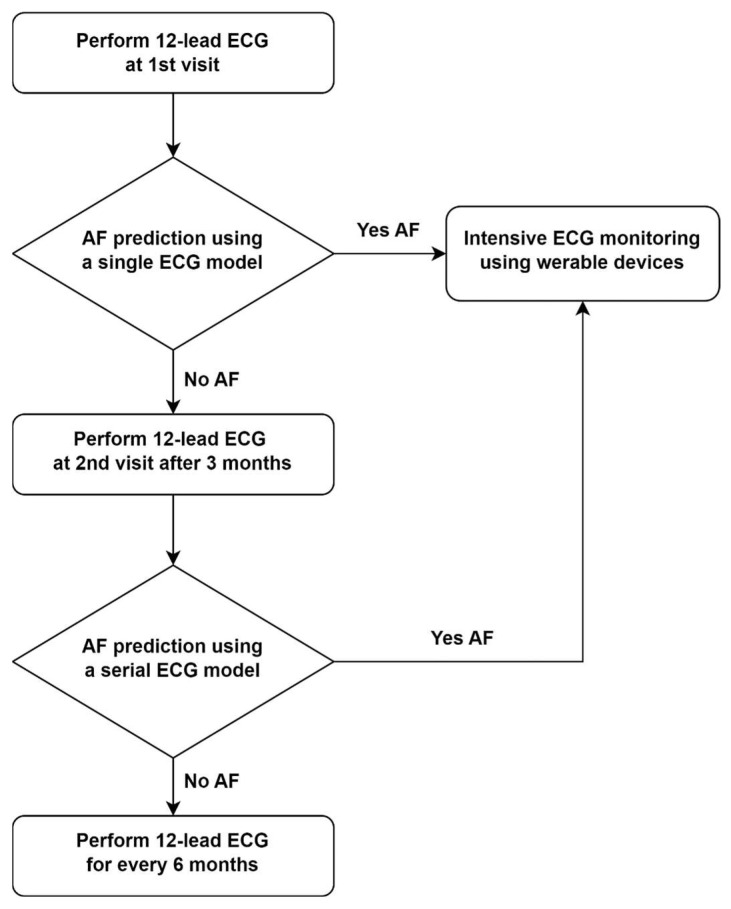
Pre-screening protocol for early diagnosis of AF.

**Figure 3 jcm-14-06675-f003:**
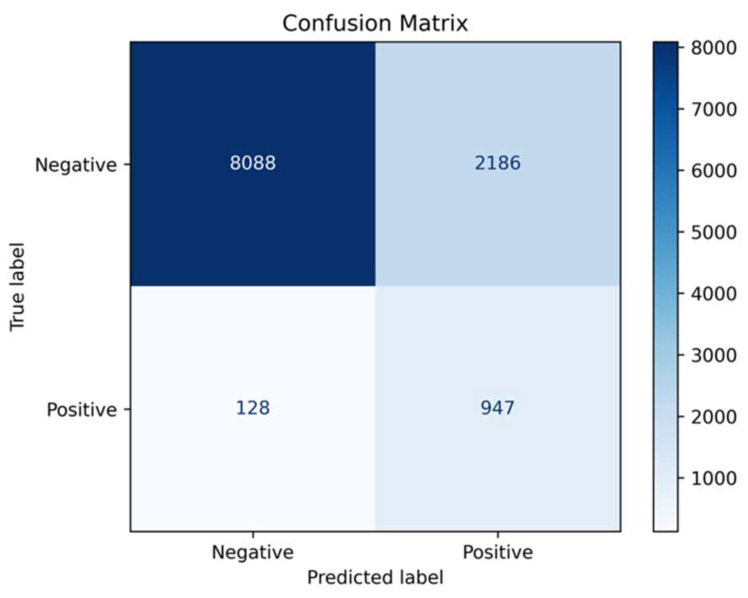
The confusion matrix to assess the performance of a classification algorithm.

**Figure 4 jcm-14-06675-f004:**
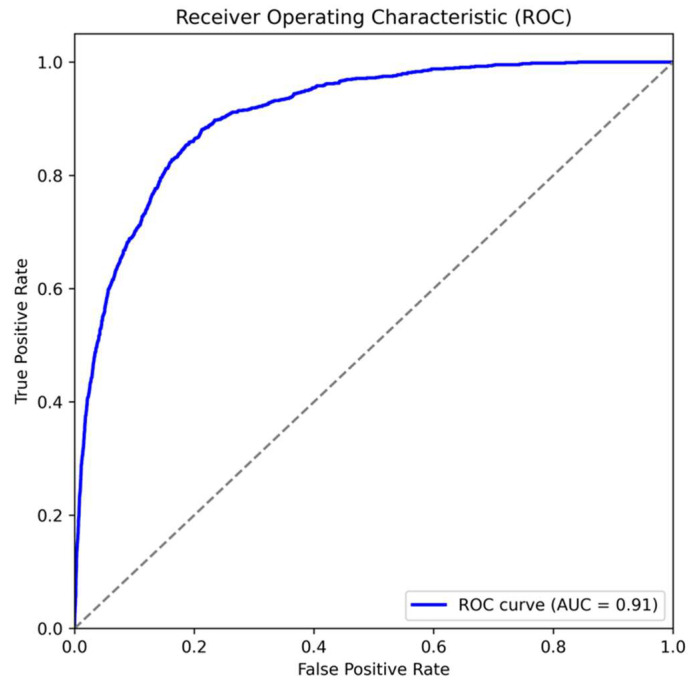
The AUROC of the proposed algorithm.

**Table 1 jcm-14-06675-t001:** Baseline characteristics.

	Overall (*n* = 164,793)	NSR Group (*n* = 154,058)	AF Group (*n* = 10,735)	*p* Value
Age, years	57.3 ± 13.7	56.5 ± 13.5	65.7 ± 13.0	<0.001
Male, *n* (%)	77,983 (47.3)	72,201 (46.9)	5782 (53.9)	<0.001
Number of ECGs per patient	2.4 ± 2.1	2.1 ± 1.6	3.5 ± 3.8	<0.001

NSR, normal sinus rhythm; AF, atrial fibrillation; ECG, electrocardiogram.

**Table 2 jcm-14-06675-t002:** Performance of the proposed algorithm.

AUROC	Sensitivity	Specificity	PPV	NPV	Accuracy	F1 Score
0.908	0.881	0.787	0.302	0.984	0.796	0.450

AUROC, area under the receiver operating characteristic curve; PPV, positive predictive value; NPV, negative predictive value.

## Data Availability

The datasets generated and/or analyzed during the current study are available from the corresponding author on reasonable request. The data are not publicly available due to institutional and ethical restrictions.

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
