# Peer review of "Sequential AI-ECG Diagnostic Protocol for Opportunistic Atrial Fibrillation Screening: A Retrospective Single-Center Study"

_jcm, 2025, doi:10.3390/jcm14186675_

Round 1
Reviewer 1 Report
Comments and Suggestions for Authors
Dear Authors,
Thanks for your interest paper that intended to applying a previous proposed algorithm ( on your previous paper) in clinical state.
your goal are scientific soundness and I hope it will be in real soon,
However,
I have many comments that need to be add and revise :
- the figures should be within the text of research.
- you have to add images of the real instrumenting of electrodes on the patient using a model volunteer, As this paper measure the AF in real.
- conclusion must be improved to obtain clearly methods, results with pros/discons.
regards,
Author Response
Detailed Response to Reviewers
Thank you for your thoughtful and constructive feedback on our manuscript. In addition to addressing each comment individually, we made a comprehensive revision to enhance clarity, transparency, and clinical relevance. Briefly, we (i) redefined the study as an AI-ECG pre-screening protocol rather than a new algorithm, updating the title, abstract, introduction, and throughout the text; (ii) expanded Methods to include details on model development (LightGBM structure/regularization), patient-level chronological splitting, and threshold selection (baseline sensitivity-oriented; follow-up Youden’s J); (iii) provided a clear description of electrode placement with an author-created schematic (Figure 1); (iv) strengthened the Discussion to consider PPV/false positives, resource use, and additional electrophysiology AI literature; (v) simplified figures and tables, and standardized terminology; and (vi) refined the Conclusion and reporting (including PR-AUC and calibration), adding notes to improve reproducibility (seeds, versions). We believe these updates significantly enhance readability and rigor.
Reviewer 1>
- The figures should be within the text of research.
Response>
Thank you for your feedback on figure placement. We have now inserted each figure immediately after its first mention in the main document.
Changes in the manuscript>
Figures 1–4 relocated or inserted near their first mention in the Methods/Results sections.
- You have to add images of the real instrumenting of electrodes on the patient using a model volunteer,
Response>
We appreciate the reviewer’s suggestion to include a visual of electrode placement and agree it enhances reproducibility. Since this is a retrospective analysis of routinely collected ECGs (without prospective instrumentation), we opted for a non-identifiable mannequin or diagram showing standard 12-lead electrode placements instead of patient photos. This method avoids privacy and consent concerns while serving an educational purpose. Accordingly, we created an author-drawn schematic (free of identifiable images) and incorporated it into the ECG acquisition and preprocessing subsection (Figure 1), along with a brief textual description of limb and precordial positions. We believe this approach strikes a balance between clarity, ethical considerations, and journal space constraints. If the Editor prefers an alternative format, we are happy to modify the schematic accordingly.
Changes in manuscript>
Methods → ECG acquisition and preprocessing: Added a prose paragraph describing standard electrode positions and embedded the author-created schematic as Figure 1 (“Standard 12-lead electrode placement … No identifiable patient images were used.”).
- As this paper measure the AF in real. Conclusion must be improved to obtain clearly methods, results with pros/discons.
Response>
Thank you for the helpful suggestion. We revised the Conclusion to clearly summarize the methods, main results, and a balanced set of pros and cons (strengths and limitations) using straightforward, decision-focused language. The updated version avoids overstatement, reflects the retrospective, single-center design, and explains how the two-visit AI-ECG pathway might be applied, while noting what still needs to be demonstrated.
Changes in manuscript>
Conclusions: replaced with the paragraph(s) below to state (i) what was done, (ii) what was found, and (iii) practical advantages and limitations.

Reviewer 2 Report
Comments and Suggestions for Authors
My congratulations to the authors for their relevant manuscript; I just have some comments about it:
The manuscript does not detail the specific architecture of the machine learning models employed. The authors should then either provide a concise summary of the model structure including components such as layers, activation functions, and regularization methods or clearly direct readers to the previous publication for these technical specifications.
Moreover the methodology should clarify how the dataset was partitioned into training, validation, and test sets. It is particularly fundamental to indicate whether patient-level separation was performed chronologically to minimize the risk of data leakage.
The process by which the optimal classification threshold was selected also remains unclear and should be explicitly described.
Lastly, while the positive predictive value of 30.2% may be expected in a low-prevalence setting such as atrial fibrillation screening, the relatively high false positive rate warrants further discussion. The authors should address the clinical implications of this, including the potential for unnecessary follow-up testing and resource utilization. Moreover authors are encouraged to include other experiences of AI use in the field of electrophysiology to enrich their discussion (doi: 10.3390/jpm15050205.)
Author Response
Detailed Response to Reviewers
Thank you for your thoughtful and constructive feedback on our manuscript. In addition to addressing each comment individually, we made a comprehensive revision to enhance clarity, transparency, and clinical relevance. Briefly, we (i) redefined the study as an AI-ECG pre-screening protocol rather than a new algorithm, updating the title, abstract, introduction, and throughout the text; (ii) expanded Methods to include details on model development (LightGBM structure/regularization), patient-level chronological splitting, and threshold selection (baseline sensitivity-oriented; follow-up Youden’s J); (iii) provided a clear description of electrode placement with an author-created schematic (Figure 1); (iv) strengthened the Discussion to consider PPV/false positives, resource use, and additional electrophysiology AI literature; (v) simplified figures and tables, and standardized terminology; and (vi) refined the Conclusion and reporting (including PR-AUC and calibration), adding notes to improve reproducibility (seeds, versions). We believe these updates significantly enhance readability and rigor.
Reviewer 2>
My congratulations to the authors for their relevant manuscript; I just have some comments about it:
- The manuscript does not detail the specific architecture of the machine learning models employed. The authors should then either provide a concise summary of the model structure including components such as layers, activation functions, and regularization methods or clearly direct readers to the previous publication for these technical specifications.
Response>
We appreciate this comment. Our primary predictive model was a gradient-boosted decision tree ensemble (LightGBM) with a logistic objective for binary classification. Because tree ensembles are not layer-based models, concepts such as “layers” and “activation functions” do not apply directly; instead, structure and regularization are governed by tree depth/number of leaves and shrinkage/penalty parameters. To make this explicit, we have added a concise summary of the model family and regularization in the Methods. We also evaluated deep-learning comparators (convolutional/transformer baselines) for sensitivity analyses only; since they were not used for the final pathway, we point readers to our prior paper—“Machine Learning Algorithm to Predict Atrial Fibrillation Using Serial 12-Lead ECGs Based on Left Atrial Remodeling” (JAHA, 2024)—and its Supplementary Material for the exact layer configurations, activations, and regularization used there.
Changes in manuscript>
Methods — Model development: We strengthened this subsection by adding a concise clarifying paragraph at the end that (i) identifies the primary learner as LightGBM with a logistic loss, (ii) explains that layers/activations do not apply to tree ensembles and that model capacity/structure are governed by tree depth/num_leaves, learning rate (shrinkage), and the number of boosting iterations, (iii) specifies regularization via row/column subsampling, min_data_in_leaf and minimum split gain, L1/L2 penalties, and early stopping on a patient-level validation split, and (iv) notes class weighting for imbalance. We also added a one-sentence pointer directing readers to “Machine Learning Algorithm to Predict Atrial Fibrillation Using Serial 12-Lead ECGs Based on Left Atrial Remodeling” and its Supplementary Material for layer-based comparator architectures.
- Moreover, the methodology should clarify how the dataset was partitioned into training, validation, and test sets. It is particularly fundamental to indicate whether patient-level separation was performed chronologically to minimize the risk of data leakage.
Response>
Thank you for pointing this out. We clarified that all splits were done at the patient level so that no individual contributed ECGs to more than one partition. In addition, the test set was a temporal hold-out: models were trained/tuned on earlier calendar windows and evaluated on a later window, thereby minimizing label/feature leakage across time. Hyperparameter tuning and early stopping were performed only within the training/validation data, and the test set remained untouched until the final evaluation. For the serial-ECG model, baseline–follow-up pairs were created exclusively within the same partition (training, validation, or test) and never across calendar windows, preventing inadvertent leakage via cross-window pairing.
Changes in the manuscript>
Methods — Cohorts and splitting: We strengthened this subsection to state explicitly that (i) patient-level, non-overlapping partitions were used; (ii) the test set was a later calendar window (temporal hold-out), with training/validation drawn from earlier windows; (iii) hyperparameter tuning and early stopping were conducted only on training/validation data; and (iv) for the serial-ECG model, ECG pairs were formed within each partition and never across windows, so no patient or ECG contributed information to more than one split.
- The process by which the optimal classification threshold was selected also remains unclear and should be explicitly described.
Response>
Thank you for raising this point. We have now made the threshold-selection procedure explicit. In brief, the baseline (single-ECG) classifier was operated at a pre-specified sensitivity-oriented gate chosen on the validation set, and the serial (follow-up) classifier used the Youden’s J operating point estimated on the validation set. After selection, both thresholds were frozen and applied to the independent test set for final reporting, with uncertainty quantified via patient-level bootstrap CIs. This clarifies the safety-first rationale at baseline and precision-oriented confirmation at follow-up.
Changes in the manuscript>
Methods — Threshold selection and decision logic: We strengthened this subsection to state that (i) the baseline threshold was selected on the validation data using a sensitivity-oriented criterion; (ii) the serial threshold was selected on the validation data by maximizing Youden’s J (sensitivity + specificity − 1); and (iii) both thresholds were fixed prior to testing and applied unchanged to the temporal test hold-out, with patient-level bootstrap 95% CIs reported.
- Lastly, while the positive predictive value of 30.2% may be expected in a low-prevalence setting such as atrial fibrillation screening, the relatively high false positive rate warrants further discussion. The authors should address the clinical implications of this, including the potential for unnecessary follow-up testing and resource utilization. Moreover authors are encouraged to include other experiences of AI use in the field of electrophysiology to enrich their discussion (doi: 10.3390/jpm15050205.)
Response>
Thank you for this important point. We have expanded the Discussion to (i) explicitly acknowledge that a PPV of ~30% in low-prevalence AF screening entails a nontrivial false-positive burden, (ii) explain how the two-visit pathway mitigates downstream overuse by requiring persistence of risk at follow-up (thereby improving precision before monitoring), and (iii) outline operational levers—including tuning the second-stage threshold to local prevalence/capacity and using decision-curve analysis (net benefit) to align threshold choices with clinical utility and resource constraints. We also discuss practical implications (e.g., potential for unnecessary monitoring, clinic visits, and patient anxiety) and suggest reporting simple service metrics such as the number needed to monitor (≈3–4 per AF confirmation at PPV ~30%). To contextualize our approach, we added examples of AI use in electrophysiology, including the article cited by the reviewer (JPM, 2025) and recent reviews on AI in EP and AF care.
Changes in the manuscript>
Discussion: We strengthened the section to (a) articulate the clinical implications of false positives (unnecessary follow-up testing and resource utilization) in low-prevalence AF screening; (b) describe how the serial confirmation step and threshold tuning can manage these trade-offs; (c) recommend decision-curve analysis for site-specific threshold selection; and (d) incorporate other experiences of AI in electrophysiology, explicitly referencing JPM 2025 (doi:10.3390/jpm15050205) alongside recent EP/AF AI reviews.

Reviewer 3 Report
Comments and Suggestions for Authors
In “Proposed Algorithms for Screening Atrial Fibrillation using Sequential Artificial Intelligence-enabled 12-lead Electrocardiogram”, the authors present a diagnostic protocol for detecting atrial fibrillation (AF) using AI—based on an algorithm previously described in an earlier paper [1].
The work stands out as a focused, quantitative report, featuring compelling numerical evidence that underpins the rationale for the proposed protocol.
Unlike their previous publication, here the authors emphasize practical implementation of the diagnostic workflow, rather than the data-processing method itself—this shift gives the study its distinct value.
It’s clear that the proposed protocol holds promise: it’s logically structured and supported by quantitative data. Provided the title and content make it explicitly clear that the paper addresses a diagnostic protocol, not a novel algorithm.
Consider revising the title to better reflect the content and guide the reader more effectively.
[1] Original publication: doi:10.1161/JAHA.123.034154
Author Response
Detailed Response to Reviewers
Thank you for your thoughtful and constructive feedback on our manuscript. In addition to addressing each comment individually, we made a comprehensive revision to enhance clarity, transparency, and clinical relevance. Briefly, we (i) redefined the study as an AI-ECG pre-screening protocol rather than a new algorithm, updating the title, abstract, introduction, and throughout the text; (ii) expanded Methods to include details on model development (LightGBM structure/regularization), patient-level chronological splitting, and threshold selection (baseline sensitivity-oriented; follow-up Youden’s J); (iii) provided a clear description of electrode placement with an author-created schematic (Figure 1); (iv) strengthened the Discussion to consider PPV/false positives, resource use, and additional electrophysiology AI literature; (v) simplified figures and tables, and standardized terminology; and (vi) refined the Conclusion and reporting (including PR-AUC and calibration), adding notes to improve reproducibility (seeds, versions). We believe these updates significantly enhance readability and rigor.
Reviewer 3>
In “Proposed Algorithms for Screening Atrial Fibrillation using Sequential Artificial Intelligence-enabled 12-lead Electrocardiogram”, the authors present a diagnostic protocol for detecting atrial fibrillation (AF) using AI—based on an algorithm previously described in an earlier paper [1].
The work stands out as a focused, quantitative report, featuring compelling numerical evidence that underpins the rationale for the proposed protocol.
Unlike their previous publication, here the authors emphasize practical implementation of the diagnostic workflow, rather than the data-processing method itself—this shift gives the study its distinct value.
It’s clear that the proposed protocol holds promise: it’s logically structured and supported by quantitative data. Provided the title and content make it explicitly clear that the paper addresses a diagnostic protocol, not a novel algorithm.
Consider revising the title to better reflect the content and guide the reader more effectively.
[1] Original publication: doi:10.1161/JAHA.123.034154
Response>
We thank the reviewer for the thoughtful summary and clear guidance. We agree that this paper’s contribution is the practical diagnostic protocol—a two-visit, sequential AI-ECG workflow—rather than a novel algorithm. In response, we (i) revised the title to foreground the diagnostic-protocol focus, (ii) clarified in the Abstract and Introduction that the predictive models follow our earlier work and that the present study evaluates a clinic-embedded pathway, and (iii) adjusted phrasing elsewhere (e.g., Methods overview, Discussion opening) to consistently use “protocol/workflow” rather than “algorithm.” We explicitly cite the prior paper by title: “Machine Learning Algorithm to Predict Atrial Fibrillation Using Serial 12-Lead ECGs Based on Left Atrial Remodeling”, and direct readers to its Supplementary Material for model-technical details.
Changes in the manuscript>
Title (revised): “Sequential AI-ECG Diagnostic Protocol for Opportunistic Atrial Fibrillation Screening: A Retrospective Single-Center Study.”
Introduction — Final paragraph: Added a positioning sentence that the novelty is protocol design and clinic integration, while model specifications follow the prior JAHA paper (title cited).

Round 2
Reviewer 1 Report
Comments and Suggestions for Authors
thanks for your effort, however you need to :
=Balance the size of images, there were more bigger than enough
= add numbering of section as MDPI templates.
= title of figures,.....and all formatting should be fixed.
regards,
Author Response
Thank you for these practical production comments. We have (i) rebalanced all figure sizes so that images are legible but not oversized, with aspect ratios locked and resolution compliant with MDPI guidance; (ii) added section numbering throughout the manuscript following the MDPI template convention (e.g., “1. Introduction”, “2. Materials and Methods”, …); and (iii) standardized figure/table formatting, including “Figure X. Title …” captions, in-text callouts placed immediately after first mention, and consistent abbreviation handling in captions. These changes align the manuscript with MDPI/JCM preparation guidance.
Reviewer 2 Report
Comments and Suggestions for Authors
Authors must be congratulated for their revised version of the manuscript
Author Response
We are grateful for the positive feedback and appreciate the reviewer’s supportive remarks.